# Assessing the Terrain Gradient Effect of Landscape Ecological Risk in the Dianchi Lake Basin of China Using Geo-Information Tupu Method

**DOI:** 10.3390/ijerph19159634

**Published:** 2022-08-05

**Authors:** Qiming Wang, Kun Yang, Lixiao Li, Yanhui Zhu

**Affiliations:** 1Faculty of Geography, Yunnan Normal University, Kunming 650500, China; 2GIS Technology Research Centre of Resource and Environment in Western China, Ministry of Education, Yunnan Normal University, Kunming 650500, China

**Keywords:** landscape ecological risk, geo-information Tupu, terrain gradient, distribution index, Dianchi Lake basin

## Abstract

The assessment of landscape ecological risk (LER) in different terrain gradients is beneficial to ecological environmental protection and risk management in different terrain gradients. Due to the impact of urban expansion, the landscape pattern of the Dianchi Lake basin (DLB) changed obviously, resulting in significant spatial difference of LER. At present, the LER assessment of the DLB is not clear, and the evolution mechanism of LER in different terrain gradients has not been revealed. Based on the LER assessment model, the geo-information Tupu method, the terrain niche gradient, and distribution index, this paper analyzed the LER and its terrain gradient effect in the DLB of China. The conclusions are as follows: (1) Since 1995, the land use type has mainly changed from grassland and cultivated land to construction land in the DLB of China. (2) The LERs in the DLB of China were mainly low, med low, and med high due to the transformation of land use type. The dominance distribution of the low and high LER was obviously constrained by terrain gradient. While the dominance distribution of med-low LER expanded to med-high terrain gradient, the dominance distribution of the med-high LER decreased to med-low terrain gradient. (3) The Tupu LERs were mainly a stable type of “medium” risk and anaphase change type of “med-high to medium” risk. The dominant distribution regions of the stable type, the prophase change type, and the continuous change type were relatively stable; the anaphase and middle change type expanded to the higher terrain gradient, and the repeated change type decreased to the med-high terrain gradient. In the process of ecological risk management and protection in the DLB, attention should be paid to the water area structure and LER control in med-high and high terrain gradients.

## 1. Introduction

Ecological risk refers to the possible combination of threats and harms to components and structures of ecosystems caused by natural or human interference directly or indirectly [1,2,3,4], specifically manifested in the reduction in health, productivity, genetic structure, economic value, and aesthetic value of the ecosystem itself [5], thus creating the probability and likelihood of adverse ecological effects [6]. Scientific management of ecological risk is an important prerequisite for the coordination between the structure and function of each component of the ecosystem and environment, but ecological risk management must be based on a scientific and accurate assessment of regional ecological risks [3,7]. Ecological risk assessment is the process of assessing the likelihood of adverse impacts on the structure and function of ecosystems, including the results of one or more stresses associated with human activities [8,9,10]. In the late 1980s, the risk assessment team of the Oak Ridge National Laboratory in Tennessee published a research paper on how ecological processes are affected by toxicology, establishing the research foundation for ecological risk assessment [11]. The US National Environmental Protection Agency completed the world’s first ecological risk assessment framework in 1992 and formed the ecological risk assessment guidelines in 1998, making the ecological risk assessment become a quantitative and universal tool for ecological risk management and decision making [12,13]. As the scope and extent of the impact of human activities on the structure and function of ecological systems continue to expand and deepen, the subjects of research have changed from a single risk to multi-risk interaction, and the scope of research has expanded from a single population, community, to an ecosystem, watershed and landscape scale [11,14,15,16].

As a new branch of regional ecological risk assessment [17], the landscape ecological risk assessment pays more attention to the impact of landscape patterns and spatiotemporal heterogeneity on ecological processes [18,19], which evaluates the impact degree and cumulative effects of natural disasters and human activities on the composition, structure, function, and process of the landscape from the perspective of disturbance mechanisms, aiming to evaluate the disturbance response of landscape to intrinsic risk sources and external human activity [20,21,22]. As the coupling result of natural factors and human activities, land use/cover changes (LUCC) are the most obvious landscape symbol on the earth’s surface and the most appropriate scale to study the impact of human activities on the ecological environment [23]. The LER change can be regarded as a comprehensive disturbance of LUCC. Therefore, the LER assessment based on LUCC effectively reflects the internal and external disturbance of landscape pattern and ecological process, and it is an important tool for assessing and managing regional ecological risk [5,23,24].

In recent years, with the development of LER research, the LER model mainly adopts the linear combination of landscape index and landscape vulnerability to evaluate the landscape risk unit. Relevant scholars analyzed the spatiotemporal changes of LER in the Shandong Peninsula, Delhi of India, Koshi River basin, and Poyang Lake region of China, which provide some scientific basis for local ecological risk management and decision support [3,5,23,25]. However, the internal process and the spatiotemporal evolution trajectory of LER are not yet clear. Geo-information Tupu can well express the spatial properties process distribution pattern of geographical things or phenomena and use the Tupu unit to visualize the characteristics of spatial atlas and temporal spectrum [26], better understand the spatial and temporal evolution mechanism of LER, and provide more accurate theoretical and practical guidance for regional ecological risk management and decision making.

Terrain, as an important physical geographic factor, affects the physical and ecological conditions of the surface [27]. Different terrains can form different combinations of water and heat, soil vertical zonality, and interference mechanisms, which further affect the evolution of ecological processes and landscape patterns [28,29,30,31]. Terrain factors (altitude, slope, terrain niche) are often used to analyze the ecosystems’ spatial structures and service functions [32,33], but there are few analyses of LER changes caused by terrain gradient. The elevation of DLB ranges from 1753 m to 2825 m, with large differences, which is conducive to the study of LER changes along the terrain gradient. The favorable climate and superior ecological environment pushed people toward agglomeration, urban sprawl, and transportation development. Wetland, cultivated land, and forest land in the center of the DLB are occupied by construction land, forcing people to move to the periphery of the city and higher terrain gradient, causing the structure, ecosystem function, and process of change on the terrain gradient, leading to the terrain gradient variation of LER. It is necessary to evaluate the LER distribution pattern and analyze its spatiotemporal evolution trajectory on the different terrain gradients. Therefore, the main objectives of this study were to (1) reveal the spatiotemporal evolution trajectory of LER with geo-information Tupu method and (2) use the terrain niche gradient and distribution index to analyze the terrain gradient effect of LER level and LER Tupu change in the Dianchi Lake basin. Our research provides a reference for promoting the sustainable development of the ecological environment of the DLB, China, and evaluating the LER and ecological environment protection in other basins.

## 2. Materials and Methods

### 2.1. Study Area

The DLB is located in the middle east part of the Yunnan Province, China. Located in 102°29′ E–103°1′ E, 24°29′ N–25°28′ N is the core area of Kunming City, including most of the 6 districts (Wuhua District, Panlong District, Guandu District, Xishan District, Chenggong District, and Jinning District), as well as parts of 2 surrounding counties (Songming County, Xundian County) (Figure 1). The region belongs to the subtropical low-latitude plateau mountain monsoon climate, with a mean annual temperature in the basin of 15 °C and mean annual precipitation of 1000 mm [34]. Kunming City is known as the “spring city” due to its pleasant climate. The area of the DLB is 2906 km^2^, accounting for 0.74 percent of Yunnan Province’s total area, but it is the most densely populated and economically developed region in the Yunnan Province, China. With the implementation of the “One Belt and One Road” Initiative, Yunnan Province has become a radiation center facing southeast Asia [35], which further promotes the rapid development of the social economy in the DLB. However, certain ecological and environmental problems have also occurred during the development of social economies. For example, with the increase in population and the enhancement and expansion of social and economic activities, a large amount of cultivated land and grassland have turned into construction land, changed the fragmentation and vulnerability of the basin patches, and changed the temporal and spatial changes of LER. Due to the obvious elevation difference (1753–2858 m) in the DLB, the development and utilization of the ecological environment by human social and economic activities are constrained by terrain gradient, which makes the LER show vertical difference along the different terrain gradients.

### 2.2. Data Source

Land use data in the DLB included 6 periods, namely 1995, 2000, 2005, 2010, 2015, 2018, which were obtained from the Data Center of Geographic Sciences and Resources and Environmental Sciences, Chinese Academy of Sciences (https://www.resdc.cn/, accessed on 1 December 2019). According to the first-level classification standard of land use/cover change, land use data are divided into cultivated land, forest land, grassland, water area, and construction land. The interpretation signs of each land use type are established based on Landsat-TM and Landsat-8 remote-sensing images. Artificial visual interpretation is used to interpret the remote-sensing data with a resolution of 30 m. The overall accuracy is above 85% [36], which meets the experimental requirements. DEM data were obtained from the Geospatial Data Cloud Platform of the Computer Network Information Center of the Chinese Academy of Sciences (http://www.gscloud.cn/, accessed on 1 December 2019). In order to make the experiment operable and computable, DEM data and land use data were unified into the same projection coordinate system and resolution. That is, the projection coordinate system was WGS_1984_UTM_ Zone_48N with a resolution of 30 m.

### 2.3. Methods

#### 2.3.1. Grid Division of Landscape Ecological Risk

In order to better show the spatial pattern of LER, it is necessary to divide the DLB of China into LER assessment units. According to the existing studies, the assessment unit area should be 2–5 times of the average patch area [37]. In this research, according to the actual situation of the DLB, China, the watershed is divided into 396 assessment units according to 3 km × 3 km grid units, and the LER value of each assessment unit center is calculated by the Fragstats4.2 software and the LER model.

#### 2.3.2. Landscape Ecological Risk Assessment Model

The LER assessment model can reflect the degree of disturbance of human social and economic activities to the structure and function of the ecosystem, and it is constructed by land use type area, landscape disturbance degree, and landscape vulnerability index [5,31,38]. The formula is as follows:(1)$$LER_{k}=\sum_{i=1}^{n}\frac{A_{ki}}{A_{k}}\times \sqrt{D_{i}\times V_{i}}$$
where *LER_K_* represents the *LER* index of the evaluation unit *k*; *A_ki_* represents the area of the land use type *i* of the evaluation unit *k*; *A_k_* represents the total area of the evaluation unit *k*; and *D_i_* represents the landscape disturbance degree of the land use type *i*. *V_i_* represents the vulnerability index of land use type *i*. As suggested in previous studies [23,24] and combined with the specific situation of the DLB of China, the vulnerability values of five land use types affected by external disturbance are different, from low to high, namely, construction land is 1, forest land is 2, grassland is 3, cultivated land is 4, water area is 5. After normalization, the vulnerability indices of each land use type are obtained.

Landscape disturbance degree is obtained by linear weighting of landscape fragmentation, landscape separation, and the landscape fractal dimension index. The calculation formula [39] is as follows:(2)$$C_{i}=\frac{n_{i}}{A_{i}}$$
(3)$$N_{i}=\frac{1}{2}\sqrt{\frac{n_{i}}{A}}\times \frac{A}{A_{i}}$$
(4)$$F_{i}=\frac{2\times \ln\left(\frac{P_{i}}{4}\right)}{\ln\left(A_{i}\right)}$$
(5)$$D_{i}=aC_{i}+bN_{i}+cF_{i}$$
where *C_i_* represents the fragmentation index of the land use type *i*, and its higher value indicates a lower stability of its landscape ecosystem system; *N_i_* represents the landscape separation index of the land use type *i*, and its higher value indicates a higher degree of landscape fragmentation. *F_i_* represents the landscape dimension index of the land use type *i*, with its value range 1–2; the higher the value, the more complex the shape of the landscape patch. *n_i_* is the number of patches of the land use type *i*. *A_i_* is the area of the land use type *i*. *A* is the total area of the land use type. *P_i_* is the perimeter of the land use type *i*; *a*, *b*, *c* represent the weights of landscape fragmentation index, landscape separation index, and landscape dimension index, respectively, and *a* + *b* + *c* = 1. Based on previous studies [3,18,40], combined with the actual situation of the DLB of China, *a*, *b,* and *c* are assigned with values of 0.5, 0.3, and 0.2, respectively.

#### 2.3.3. Geo-Information Tupu Method

Geo-information Tupu can combine the “attribute-pattern-process” information of the LER value [41], effectively integrate the spatiotemporal variation trajectory of LER, and better understand the mechanism of LER change. In this paper, geo-information Tupu is proposed to enrich the research scope of the LER assessment. The computational method [42] is summarized as follows:(6)$$TP=\sum_{i=1}^{n}G_{i}\times 10^{n-i}\left(n\geq 2\right)$$
where *TP* represents the unit attribute value of the LER change Tupu in successive periods; *i* is a serial number of different periods; *n* is the total number of consecutive periods; *G_i_* is the codes of the level of LER at the ith time node of a given pixel in chronological order. According to the natural breakpoint method, the LER level in each time node can be classified into five levels: low (L), medium low (ML), medium (M), medium high (MH), and high (H). In this study, the 1995–2018 six periods of six different LER levels of coupling form a six-figure coding pattern of LER change Tupu, a Tupu unit order consistent with the LER level time series, namely, the first code on behalf of the LER level in 1995, the bottom on behalf of the LER level in 2018.

According to the geo-information Tupu method, we superimposed different time nodes Tupu units to form the combination of different LER level units in the study area. According to the change trajectory of LER level at different time nodes, the change trajectory of LER level can be divided into the following six LER change Tupu types. (1) Stable type: the level of LER remains the same and does not change in all time nodes. For example, the trajectory unit values of LER level such as L-L-L-L-L-L, ML-ML-ML-ML-ML-ML, M-M-M-M-M-M, etc., are considered to be of the stable type. (2) Prophase change type: the level of LER changed for at least one time node between 1995 and 2000 and remained unchanged at other time nodes. For example, the trajectory unit values of LER level such as L-ML-L-L-L-L, ML-ML-L-L-L-L, ML-L-L-L-L-L, etc., are considered to be of the prophase change type. (3) Middle change type: the level of LER changed for at least one time node between 2005 and 2010 and remained unchanged at other time nodes. For example, the trajectory unit values of LER level such as L-L-L-ML-L-L, L-L-ML-ML-L-L, L-L-ML-L-L-L, etc., are considered to be of the middle change type. (4) Anaphase change type: the level of LER changed for at least one time node between 2015 and 2018 and remained unchanged at other time nodes. For example, the trajectory unit values of LER level such as L-L-L-L-ML-L, L-L-L-L-ML-ML, L-L-L-L-L-ML, etc., are considered to be of the anaphase change type. (5) Repeated change type: the level of LER was the same at time nodes between 1995 and 2018 but changed at other time nodes. For example, the trajectory unit values of LER level such as L-ML-H-M-MH-L, L-ML-L-M-MH-L, L-ML-L-M-ML-L, etc., are considered to be of the repeated change type. (6) Continuous change type: the level of LER was different at time nodes between 1995 and 2018, but it changed 4–5 times at all time nodes. For example, the trajectory unit values of LER level such as L-ML-M-MH-H-ML, L-ML-L-ML-M-ML, L-ML-L-L-M-LM, etc., are considered to be of the continuous change type.

#### 2.3.4. Terrain Niche Index

In order to better reflect the distribution pattern of LER on different terrain gradients, we merged the DEM and slope in the ArcGIS10.7 software to establish the terrain niche index (TNI) [31] (Figure 2). As a comprehensive terrain factor, it can make up for the non-obvious changes in low terrain gradient caused by high elevation, small slope, or low elevation and large slope, and effectively reflect the spatial differentiation of terrain gradient [43]. The calculation [31] is as follows:(7)$$TNI=\ln\left[\left(\frac{E}{\bar{E}}+1\right)\times \left(\frac{S}{\bar{S}}+1\right)\right]$$
where *TNI* represents the terrain niche index; E and $\bar{E}$, respectively, represent the altitude value of any point and its average altitude value around the neighborhood in the research around; S and $\bar{S}$ are the slope value of any point and its average slope value around the neighborhood in the study area, respectively. The higher the altitude value and the higher the slope value, the larger the terrain niche index, and vice versa [27].

#### 2.3.5. Distribution Index

The distribution index (*DI*) reflects the distribution degree of LER (LER change Tupu) in different terrain niche gradients [27], which effectively eliminates the effect of terrain gradients and the difference of LER (LER change Tupu) areas. The formula [32] is as follows:(8)$$DI=\left(T_{ij}/T_{i}\right)/\left(T_{j}/TA\right)$$
where *DI* is the distribution index of different LER levels (LER change Tupu); *i* is the LER level (LER change Tupu), *j* is the terrain niche gradient factor; *T_i_* is the total area of LER level (LER change Tupu) *i*. *T_ij_* is the total area of LER level (LER change Tupu) *i* on terrain niche gradient *j*, *T_j_* is the total area of terrain niche gradient *j*; *TA* is the total area of the study area. *DI* > 1 indicates that a certain LER level (LER change Tupu) presents a dominant distribution on a certain terrain niche gradient, and the larger the value, the more obvious the dominance degree; *DI* = 1 indicates that a LER level (LER change Tupu) is evenly distributed on a certain level of terrain niche gradient; *DI* < 1 indicates that a certain LER level (LER change Tupu) presents a disadvantage distribution on a certain terrain niche gradient, and the smaller the value, the more obvious the disadvantage degree.

## 3. Results

### 3.1. Structure and Pattern Changes of Land Use in the Dianchi Lake Basin

The ratio and spatial pattern changes of land use in the DLB are shown in Figure 3. In 1995, the main land use types were forestland, cultivated land, and grassland, accounting for more than 72.47% of the total area in the DLB of China, but by 2018, the main land use types had changed to forestland, construction land, and cultivated land, accounting for 73.47% of the total area. On the whole, cultivated land and grassland decreased year by year, by 3.44% and 1.45%, respectively, while the construction land increased year by year, by 5.4%. The forestland and water area had little change. It was mainly caused by the eastward and northward sprawl and southeastward jump expansion of the city to occupy a large amount of cultivated land and grassland. As can be seen from the land use transfer matrix (Table 1), cultivated land, grassland, and forestland were less transferred in and more transferred out among land use types, and they were mainly transformed into construction land, accounting for 97.816 km^2^, 38.047 km^2^, and 20.148 km^2^, respectively, while construction land was least transferred out, accounting for only 1.127 km^2^. The agglomeration of the population to Kunming city and the expansion of urbanization changed the land use structure of the DLB, China, transformed cultivated land, grassland, and forestland into construction land, gradually changed the disturbance degree and vulnerability of the landscape structure, and finally changed the LER of DLB, China.

### 3.2. Spatial Pattern Distribution of LER in the Dianchi Lake Basin

The common Kriging method in the ArcGIS10.7 software was used to interpolate the LER values of 396 grid centers to obtain the LER values of the whole DLB. In order to obtain the spatial distribution of the LER value with obvious difference, the natural breakpoint method is adopted to classify the LER of the DLB. The level of LER in the DLB is divided into five categories: low-risk area (0.142 ≤ LER ≤ 0.176), med-low area (0.176 < LER ≤ 0.200), medium-risk area (0.200 < LER ≤ 0.218), med-high-risk area (0.218L < LER ≤ 0.237), high-risk area (LER ≥ 0.237).

The proportion of the LER level in the DLB varied greatly over time from 1995 to 2108 (Figure 4). The LER levels of the DLB were dominated by low-risk area, medium-risk area, and medium-high-risk area, accounting for more than 70% of the total risk areas, while low- and high-risk areas represented less than 30% of them. Overall, the proportion of low-risk and medium-low-risk areas increased by 4.75%, while the proportion of the medium-high-risk and high-risk areas decreased by 8.23%, which also reflecting the gradual decrease in the LER level in the DLB, indicating that the overall LER status gradually increased in the DLB.

The spatial distribution of LER was significantly different in the DLB (Figure 4). The high-risk area was mainly distributed around the Dianchi Lake and around the northeast border of the DLB, and its range and proportion were decreasing gradually. The low-risk area was mainly distributed in the dense urban areas in the DLB, whose scope and proportion were gradually increasing with the urban sprawl and expansion. The medium-high-risk area was mainly distributed in the south of the DLB, and its range was gradually decreasing. The medium-risk area was mainly distributed in the north and southeast of the DLB, and its range and proportion were increasing in the southeast. The medium-low-risk area was mainly distributed in the periphery of the low-risk area in a circular distribution, and its range was gradually decreasing.

### 3.3. LER Spatiotemporal Evolution Trajectory in the Dianchi Lake Basin

The Raster Calculator tool of Spatial Analyst in the ArcGIS10.7 software was used to overlay and classify the LER levels of six periods from 1995 to 2018 using the Geo-information Tupu method to obtain the LER spatiotemporal variation trajectory (Figure 5). A statistical analysis of the area and proportion of the Tupu types (Table 2) is helpful to analyze the changes of the structure and quantity of different LER change Tupu types. The LER change Tupu of DLB was dominated by the stable type and anaphase change type, covering 2118.93 km^2^ and 388.85 km^2^, respectively, accounting for 86.59% of the total area of the Tupu change type. They were widely distributed in the DLB in patches and strips, indicating that the landscape ecological risk system of DLB is relatively stable. This is beneficial to the sustainable development of DLB.

The stable type was distributed in the whole DLB, accounting for more than 70% of the total area, among which the “medium-medium-medium-medium-medium-medium” risk was the maximum Tupu type, accounting for 37.70% of the total area of the stable type. DLB was located in the region of rapid urbanization and population concentration. Frequent social and economic activities continued to disturb the change of land use types, which made the structure and function of the landscape ecosystem continuously disturbed. Therefore, the LER of the stable type was mainly the stability of medium risk.

The anaphase change type was mainly distributed around the Dianchi Lake, the periphery of the main urban area and the south of the DLB, accounting for 13.43% of the total area, among which the “medium high-medium high-medium high-medium high-medium-medium” risk was the maximum Tupu type, accounting for 40.18% of the total area of the anaphase change type. Mainly due to the construction of the ecological environment around the Dianchi Lake, the further implementation of returning farmland to grassland in the southern part of DLB, and the southward expansion of urbanization, the anti-interference ability and stability of the landscape ecosystem were enhanced. Therefore, the LER of the anaphase change type was reduced.

The prophase change type was mainly distributed around the main urban area, and a few bands were distributed in the central and southern part of the DLB, accounting for 4.19% of the total area, of which the “medium low-medium low-low-low-low-low” risk was the maximum Tupu type, accounting for 40.27% of the total area of the prophase change type. It was mainly because of the city expansion in the early stage of urbanization that the proportion of construction land increased; the anti-interference ability of landscape ecology was strong. Therefore, the LER of the prophase change type was reduced.

The continuous change type was mainly distributed around the prophase change type and a small amount in the south of the DLB, accounting for 8.65% of the total area, among which the “medium low-medium low-medium low-low-low-low” risk was the maximum Tupu type, accounting for 47.75% of the total area of the continuous change type. Due to the continuous expansion of urbanization, construction land occupied a large amount of cultivated land around the city, and the strong external anti-interference ability of construction land made the LER of the continuous change Tupu gradually decrease along the urban center of the DLB.

The middle change type and repeated change type were mainly linear and strip distributed in the northern boundary and southern part of the DLB, accounting for 0.58% of the total area, among which the “medium high-medium high-medium-medium high-medium high-medium high” and “medium-medium-medium-medium high-medium high-medium” risks represented the corresponding maximum Tupu types. It was the result of a small amount of grassland and cultivated land interconversion in the DLB.

### 3.4. Terrain Gradient Effect of LER in the Dianchi Lake Basin

The terrain niche index is calculated using the DEM and slope by the map algebra tool of the ArcGIS version 10.7 software (Redlands, CA, USA). According to the principle of equal interval and the terrain distribution characteristics of the DLB, the TNI is divided into 16 grades from low to high and grouped into four terrain niche gradients: low (1–4), mid-low (5–8), mid-high (9–12), and high (13–16). According to the distribution index, the distribution characteristics of the LER level on different terrain gradients can be calculated (Figure 6).

From 1995 to 2018, the distribution patterns of low, middle, mid-high, and high LER regions in DLB did not change significantly on the terrain gradient, but the distribution advantage of the mid-low LER regions had expanded to the higher terrain gradient, indicating that the LER level of the middle-low-risk regions decreased, and the overall LER level decreased.

As can be seen from Figure 6a, with the rising terrain gradient, the distribution index of low-risk regions firstly increased and then decreased, and its advantage distribution regions were mainly concentrated in the low, mid-low terrain gradient. In the low terrain gradient, the advantage distribution index increased, which is related to the occupation of cultivated land by construction land. The construction land has a strong anti-interference ability, and the level of LER is reduced. In the mid-low terrain gradient, the advantage distribution index decreased, mainly because the cultivated land was occupied by the construction land in the low terrain gradient, forcing the cultivated land to transfer to the mid-low terrain gradient. However, the fragmentation degree of cultivated land is high, and the LER level increased.

As can be seen from Figure 6b, with the rising terrain gradient, the distribution index of mid-low-risk regions also firstly increased and then decreased. However, the ranges of advantage distribution expanded from the low and mid-low terrain gradients in 1995 to the low, mid-low, and mid-high terrain niche gradients in 2018, and the advantage distribution index of the mid-high terrain gradient increased. The implementation of the policy of returning farmland to forest and grassland in the mid-high terrain gradient reduces the landscape fragmentation degree in the mid-high terrain gradient and reduces the LER level in the mid-high terrain gradient.

As can be seen from Figure 6c, with the rising terrain gradient, the distribution index of middle risk regions presented a distributed trend of “rapid increasing-slow reduction-increasing”, its advantage distribution areas mainly concentrated in mid-low, mid-high, high terrain gradient. In the mid-low terrain gradient, the advantage distribution index increased, and the landscape vulnerability and ecological risk increased because the grassland was converted to cultivated land. In the mid-high and high terrain gradient, the advantage distribution index decreased. Due to the implementation of the national return of farmland to forest and grassland and planting of economic forests in the middle and high terrain gradients, the LER level decreased.

As can be seen from Figure 6d, with the rising terrain gradient, the distribution index of mid-high risk regions presented a distributed trend of “increase-decrease-increase”, and its advantage distribution regions were mainly concentrated in the low, mid-low, and high terrain gradients. In the mid-low terrain gradient, the advantage distribution index decreased, which is related to the increase in construction land due to urbanization expansion. The strong anti-interference ability of the construction land leads to the reduction in the LER level. In the low and high terrain gradients, the advantage distribution index increased. Due to the large water area in the low terrain gradient and the frequent occurrence of natural disasters in the high terrain gradient, the LER level increased.

As can be seen from Figure 6e, with the rising terrain gradient, the distribution index of high-risk regions presented a distributed trend of “rapidly decreasing-stable and slowly decreasing”; its advantage distribution regions were concentrated in the first grade of the low terrain gradient. The advantage distribution index also increased from 1995 to 2108, and its value was as high as 6.33, indicating that high-risk regions had an absolute advantage in the low terrain gradient. The main reason was that the water area was distributed in the low terrain gradient, and the water area was vulnerable and disturbed, which was easy to cause floods and other natural disasters.

### 3.5. Terrain Gradient Effect of LER Change Tupu in the Dianchi Lake Basin

The distribution index of the LER change Tupu obviously existed different on the terrain gradient. In order to better reflect the directivity of LER change Tupu on different terrain gradients, the maximum change Tupu of LER should be introduced to explore the response of terrain factors and human activities to LER. As can be seen from Figure 7 and Table 3, the distribution advantage of the LER change Tupu was different in different terrain gradients, and the directivity of the distribution advantage of maximum Tupu was different on the terrain gradient.

With the increase in terrain niche gradient, the distribution index of the stable type firstly decreased and then slowly increased, and its advantage distribution regions were mainly concentrated in the lowest, med-high, and high terrain gradient. On the lowest terrain gradient, the maximum change Tupu was dominated by the “High-High-High–High-High-High” risk and had a stable distribution advantage. The main reason is that the water area was distributed on the lowest terrain gradient, which accounted for a large proportion of the area and remained stable. Moreover, the water area was strictly bound by the terrain and had a high degree of ecological risk vulnerability. On the med-high and high terrain gradient, the maximum change Tupu was dominated by the “Medium-Medium-Medium-Medium–Medium-Medium” risk and had a stable distribution advantage. Forestland was mainly distributed on this terrain gradient, which occupied a large area and had ecological functions, such as soil and water conservation. However, there were some natural disasters, such as landslides, debris flow, and soil erosion, on this terrain gradient. The interaction between the above two factors made the LER of the stable type mainly moderate.

With the increase in terrain gradient, the distribution index of the prophase change type and continuous change type firstly increased and then decreased, and their advantage distribution regions were mainly concentrated in the low and med-low terrain gradients. At the low and med-low terrain gradient, the maximum change Tupu patterns of the prophase change type and continuous change type were dominated by the “Med-low-Med-low-Med-low-Med-low-Low-Low” and “Med-low-Med-low-Med-low-Low-Low-Low” risk, respectively, and had a stable distribution advantage. With the population concentration and expansion of urbanization, human activities advanced from low to med-low terrain gradient. As a result, the contiguous cultivated land was gradually occupied by construction land from the low to med-low terrain gradient, and the contiguous construction land had a strong anti-interference ability, so that the LER level of the prophase and continuous change type changed from medium to low level.

With the increase in terrain gradient, the distribution index of the middle change type firstly increased and then tended to be stable, finally decreasing. Its advantage distribution regions were mainly concentrated in the med-low and med-high terrain niche gradient. In the med-low and med-high terrain gradient, the maximum change Tupu type of “Med-high-Med-high-Medium-Med-high-Med-high-Med-high” had a stable distribution advantage. In the early stage, large areas of cultivated land around the city were occupied by construction land. In order to balance the occupation and supplement and meet the food needs, the grassland of the med-low gradient was converted into cultivated land, which had a high degree of fragmentation and was more disturbed by human activities, and the LER level changed from med-high to high terrain gradient. In the high gradient, the maximum change Tupu type of “Medium-Medium-Medium-Med-high-Medium-Medium” appeared to present novel distribution advantages. The high terrain was suitable for the growth of forestland. Under the stimulation of economic interests, people planted a large mass of economic woods, such as walnuts, olive, and chestnuts, on high terrain, occupying a large amount of grassland, whose LER level of the middle change type reduced from med-high to middle in the high terrain gradient.

With the increase in the terrain gradient, the distribution index of the anaphase change type firstly increased and then decreased, and its advantage distribution regions were mainly concentrated in the 2–7 grade terrain gradient. The maximum change Tupu of “Med-high-Med-high-Med-high-Med-high – Medium-Medium” had a distribution advantage on the 2–11 grade terrain niche gradient. It can be found that the advantage distribution regions of the anaphase change type advanced to the med-high terrain gradient, and its ecological risk level decreased. With the continuous promotion and improvement of the national policy of returning cultivated land to forestland in the later period, the sloping farmland on the med-high terrain gradient was transformed into forestland and grassland, so that the natural ecosystem developed toward a benign direction, and the LER level decreased from med-high to medium terrain gradient.

With the increase in terrain gradient, the distribution index of the repeated change type firstly decreased and then increased wavelike, and its advantage distribution regions were mainly concentrated in the 1, 7–16 grade terrain gradient. The maximum change Tupu of the “High-Med-high-Med-high-High- High-High” risk had a distribution advantage in 1, 15–16 grade terrain gradient. In the first terrain grade, there was a large area of water, and the LER level was high; in the 15–16 grade terrain gradient, the frequent occurrence of natural disasters led to a high level of ecological risk. The maximum change Tupu of “Medium-Medium-Medium-Med-high-Med-high-Medium” had a distribution advantage in 8, 11–13 terrain niche gradient, but its advantage distribution regions narrowed. This indicates that the LER level of the repeated change type decreased in the med-high terrain gradient.

## 4. Discussion

### 4.1. Analysis of LER Change in the Dianchi Lake Basin

Landscape level is the most suitable scale to study human activities and ecological environment, while LUCC is the most basic unit of landscape and ecological environment management [6]. LUCC is closely related to regional LER, and the change of land use will inevitably lead to change of LER. During 1995–2018, major land use types changed greatly in the DLB, with a large amount of cultivated land and grassland being converted into construction land (Figure 3). However, the construction land shows low vulnerability after being disturbed by human activities during 1995–2018 (Table 4), which gradually reduced the LER of the DLB. This is mainly due to the rapid development of the built-up area of the watershed, which leads to the formation of concentrated and continuous patches of construction land, thus enhancing the aggregation and internal stability of the patches of stability of the landscape ecosystem [44]. The average value of LER changed from 0.213 in 1995 to 0.206 in 2018, indicating that the overall LER status of the DLB gradually improved. This trend is similar to the results of previous studies [5,45]. As can be seen from the spatial distribution of land use types and landscape ecological risks (Figure 3 and Figure 4), the spatial distribution of landscape ecological risks was highly coupled with the regional land use distribution in the DLB during 1995–2018.

Low-risk regions are concentrated in the main urban area dominated by construction land, and their distribution is gradually expanding. Due to the rapid expansion of built-up areas, the construction land is concentrated, and the contiguous patches are distributed, which enhances the aggregation and internal stability of patches of the landscape ecosystem. On the contrary, medium-high and high-risk areas are mainly distributed in the lake area and the northern boundary area of the DLB. On the one hand, Dianchi Lake is dominated by the water area, with a single landscape type, high fragmentation and fragility, and high ecological risk. On the other hand, due to poor soil and water conservation in the northern basin boundary, the land use mode is unreasonable and lacks management, which leads to increased landscape fragmentation and increased landscape ecological risk [46]. Since 2010, the LER of the lakeside area has been reduced, mainly due to the implementation of lakeside ecological management by the local government [40], which improved the lakeside ecological risk capacity. This also indicated that LUCC directly affected the structure and configuration of the ecosystem, leading to the distribution and structure of the LER.

### 4.2. Terrain Gradient Effect on LER and Its Change Tupu in the Dianchi Lake Basin

Terrain is an important factor affecting landscape structure and spatial pattern [27,32,47]. There are obvious vertical differences of land use type structure and pattern on the different terrain niche gradients, and land use type changes indirectly affect the changes of LER through landscape pattern and process [31]. In this paper, the terrain niche index was selected as a comprehensive natural factor affecting LER change, which could well represent the combination of intermediate terrain with high altitude and small slopes or low altitude and large slopes. High-risk regions were obviously constrained by the terrain gradient. Its distribution advantage was concentrated in the first grade terrain gradient, while the water area was mainly distributed in the first grade terrain gradient, and the water area had high vulnerability and disturbance degree, which made the LER level high. Low-risk regions were also restricted by the terrain gradient, mainly concentrated in the low and med-low terrain niche gradient. Due to urbanization expansion, large amounts of cultivated land and grassland were converted into construction land, which formed concentrated and contiguous patches, improving the agglomeration and internal stability of the ecosystem with low vulnerability. The distribution advantage of medium-risk regions was extensive, and the distribution advantage of med-low-risk regions expanded to med-high terrain gradient, which was less constrained by terrain gradient due to the continuous implementation of the policy of returning farmland to forest and grassland. The area of forest and grassland increased in the med-high terrain gradient, while the landscape fragmentation degree of the forest and grassland was smaller, and the ecological risk was lower.

Geo-information-Tupu combined with the terrain niche index can well reflect the spatiotemporal variation trajectory of LER on different gradients during continuous time nodes [48]. We can find that the distribution advantage prophase change type, the continuous change type, and the anaphase change type were mainly distributed in the low and med-low terrain gradient. Cultivated land and grassland continuously translated into construction land and were mainly distributed in the low and med-low terrain gradient. The stable type was mainly distributed in the lowest, middle-high, and high terrain gradient. Water was distributed in the first grade terrain gradient, and woodland was distributed in the med-high and high terrain gradient, and their areas were relatively stable. The middle change type expanded to the high terrain gradient; yet, the repeated change type shrank to the med-high terrain gradient. The study of the LER’s spatial-temporal variation trajectory and land use allocation pattern on different terrain gradients is beneficial for local government departments to implement precise LER management on different terrain gradients.

### 4.3. Advantages and Limitations

The innovation of this study was to provide a feasible method, that is, to use geo-information Tupu and distribution index to reflect the spatiotemporal change trajectory and vertical variation characteristics of LER. First, the ArcGIS10.7 software was used to classify LER into five risk levels (low, med-low, medium, med-high, high) according to the natural discontinuity method. The LER Tupu of the DLB was obtained by superposing LER levels in 1995–2018 using the geo-information Tupu method. It can be found that the DLB was mainly characterized by the stable type of maximum change Tupu “medium-medium-medium-medium-medium-medium” risk and the anaphase change type of maximum change Tupu “med-high-med-high-med-high-med-high-medium-medium” risk. From the above main Tupu types, it can be seen that medium-risk regions occupied the largest area. The government should optimize the ecological risk management of medium-risk regions, such as building landscape ecological corridors, lakeside ecological parks, national parks, and so on [38,49,50].

In this study, the LER assessment model was used to assess the LER change of the DLB, which provided the theoretical basis and decision support for LER assessment and management in the DLB, but there were some limitations. When calculating the landscape disturbance degree, different scholars assign different weight values to the landscape fragmentation index (*Ci*), landscape separation index (*Ni*), and landscape fractal dimension index (*Fi*). Some scholars set the weight values of the above three indices as a = 0.6, b = 0.3, and c = 0.1 [5,24], but others set the weights as a = 0.5, b = 0.3, c = 0.2 [3,18,31]. Generally speaking, the three weights are in the order of a > b > c. The expert assignment method is used to assign the landscape vulnerability index. The assignment of landscape vulnerability index adopts the expert assignment method [5]. Although the assignment has a certain scientific basis, the assignment of landscape vulnerability index has a certain subjectivity due to the regional differences of land use type, which affects the assessment accuracy of the LER value. In addition, the landscape vulnerability degree is affected by the ecosystem services, such as soil conservation, water conservation, carbon storage, water yield, nitrogen and phosphorus output [21]. Therefore, the supply and demand of the ecosystem services is used to evaluate the landscape vulnerability index of different regions, which can reasonably assess the regional LER level. Assessing the landscape vulnerability index from the perspective of ecosystem service supply and demand is one of the research directions for improving the accuracy of the LER assessment in the future.

## 5. Conclusions

Based on land use data from 1995 to 2018 and DEM data, this study analyzed the spatiotemporal change of landscape type, the spatial pattern and structure changes of LER and LER change Tupu, and their terrain gradient effect in the DLB by using the LER model, the geo-information Tupu method, the terrain niche gradient, and the terrain distribution. The main conclusions are as follows:(1)The level of LER decreased slowly in the study area. Urban expansion made the range and proportion of low-risk areas gradually increase, while the decrease in cultivated land area in the south and the strengthening of lakeside ecological construction in the study area made the range and proportion of med-high and high-risk areas shrink. The study area was dominated by the stable type of medium risk and the anaphase change type of medium-high to medium risk, which accounted for more than 85% of the total area of the study area. The LER change trajectory was relatively stable.(2)The dominant distribution of low risk and high risk was significantly constrained by the terrain gradient. With the increase in terrain gradient, the dominant distribution of medium-low risk advanced to a higher terrain gradient, which reduced the level of LER, and the dominant distribution of medium risk was widespread, and the dominant degree was reduced. The dominant distribution of med-high risk advanced to a lower terrain gradient, and its LER level decreased.(3)With the increase in terrain gradient, the distribution indices of the prophase change type, continuous change type, and anaphase change type increased first and then decreased in the study area, and the dominant distribution interval was low and med-low terrain gradient, but the maximum change Tupu had different directivity for the above three types. The LER level of the stable type was relatively stable due to the constraints of the terrain gradient. The dominant distribution interval of the middle change type was enlarged on the high terrain gradient, and the dominant distribution interval of the repeated change type was reduced on the med-high terrain gradient.

## Figures and Tables

**Figure 1 ijerph-19-09634-f001:**
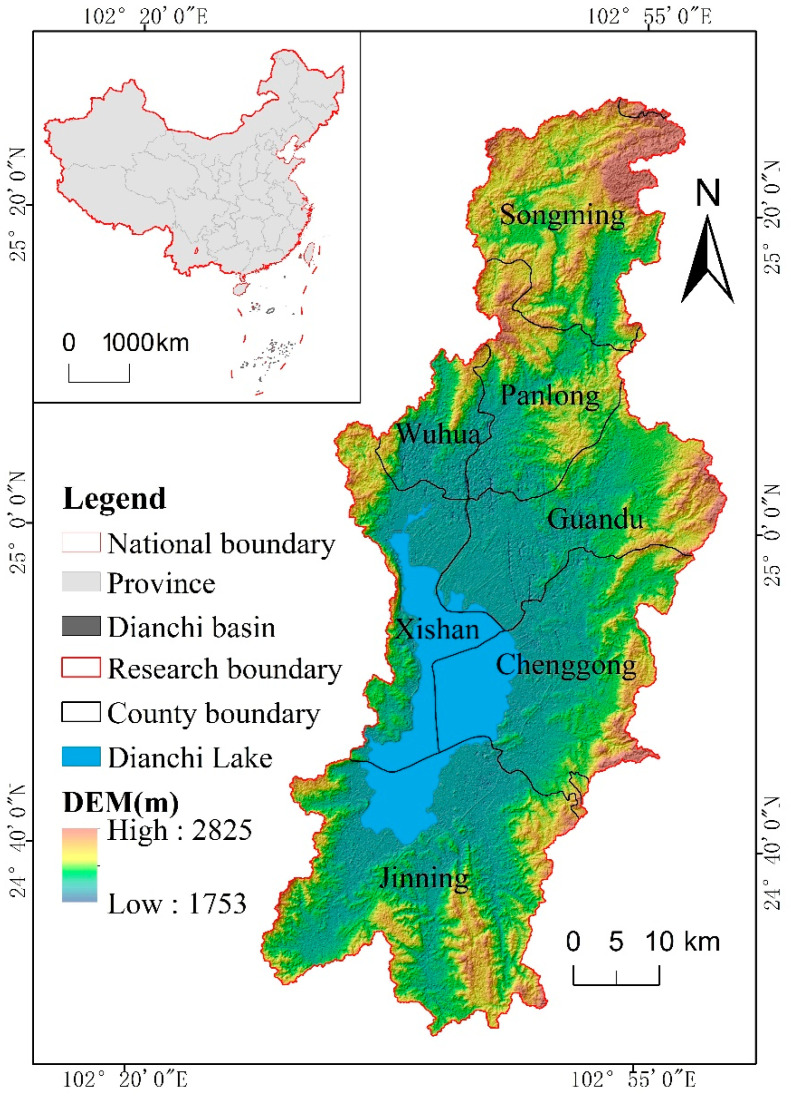
Location of the Dianchin Lake basin, China.

**Figure 2 ijerph-19-09634-f002:**
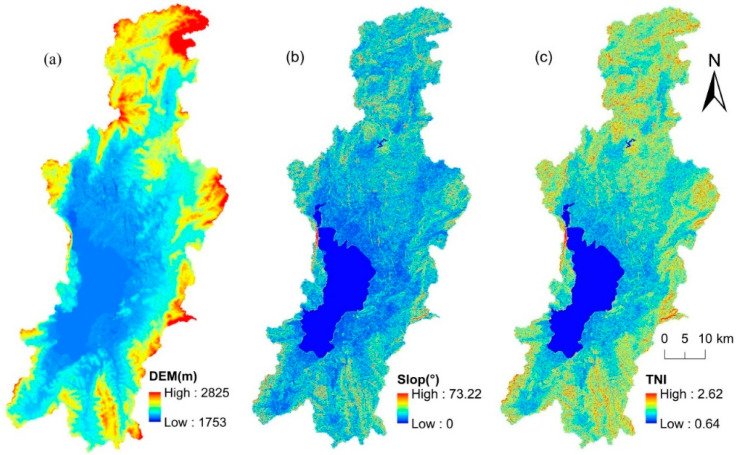
DEM (**a**), Slope (**b**), and TNI (**c**) of the Dianchi Lake basin, China.

**Figure 3 ijerph-19-09634-f003:**
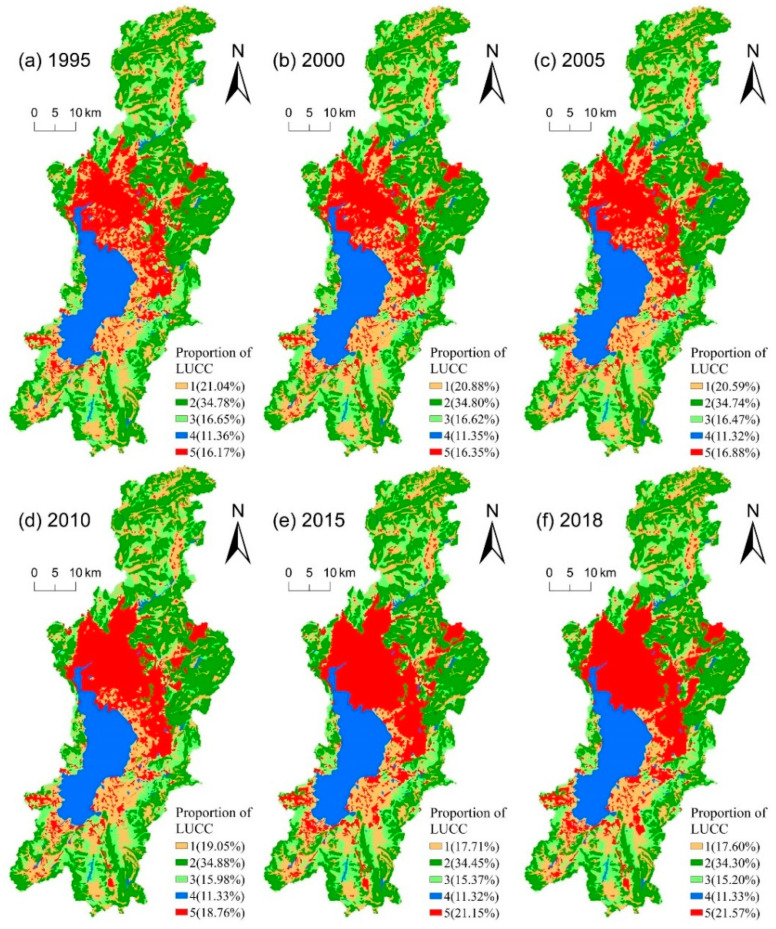
LUCC structure and distribution in the Dianchi Lake basin of China from: (**a**) 1995, (**b**) 2000, (**c**) 2005, (**d**) 2010, (**e**) 2015, (**f**) 2018. Note: 1: Cultivated land, 2: Forestland, 3: Grassland, 4: Water area, 5: Construction land.

**Figure 4 ijerph-19-09634-f004:**
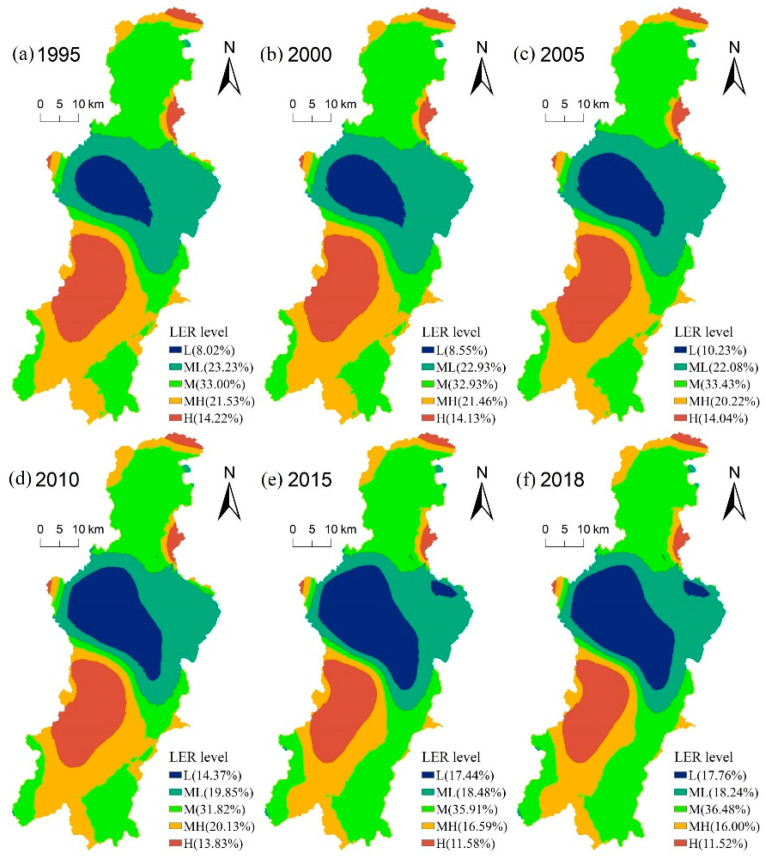
Spatial pattern and proportion changes of the LER in the Dianchi Lake basin of China in (**a**) 1995, (**b**) 2000, (**c**) 2005, (**d**) 2010, (**e**) 2015, (**f**) 2018. Note: L (low), ML (medium-low), M (medium), MH (med-high), H (high).

**Figure 5 ijerph-19-09634-f005:**
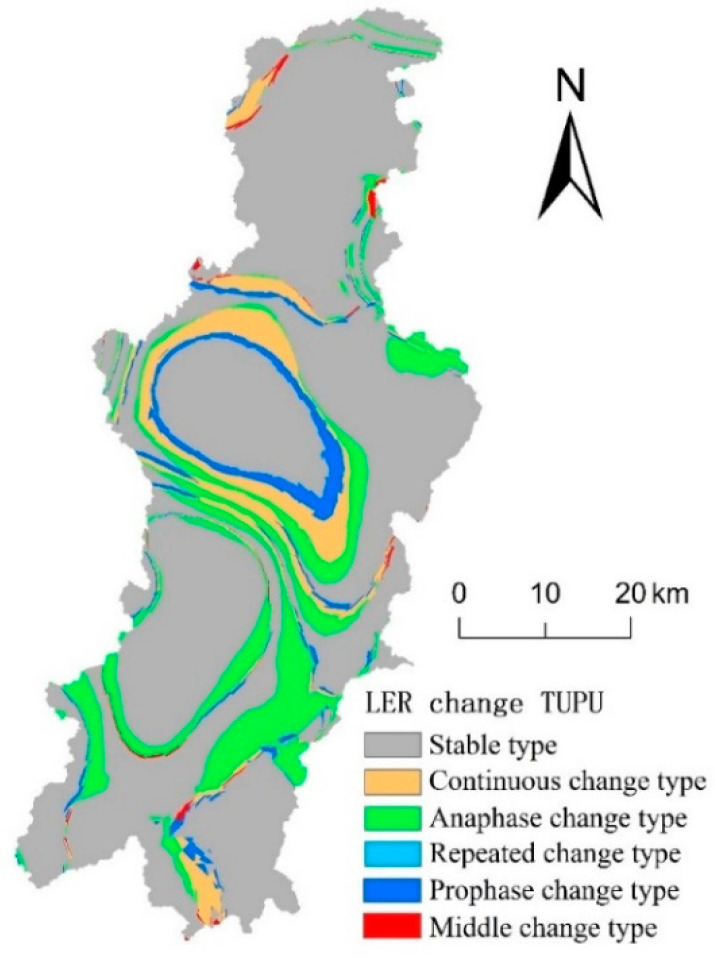
Spatial distribution of LER change Tupu in the Dianchi Lake basin.

**Figure 6 ijerph-19-09634-f006:**
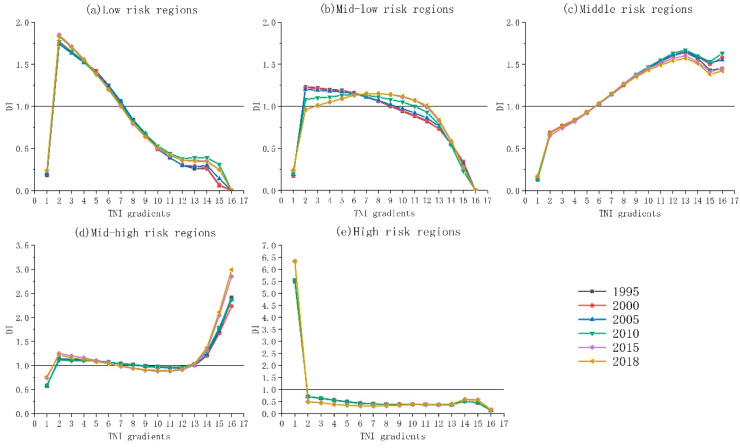
The terrain gradient distribution of LER in the Dianchi Lake basin of China.

**Figure 7 ijerph-19-09634-f007:**
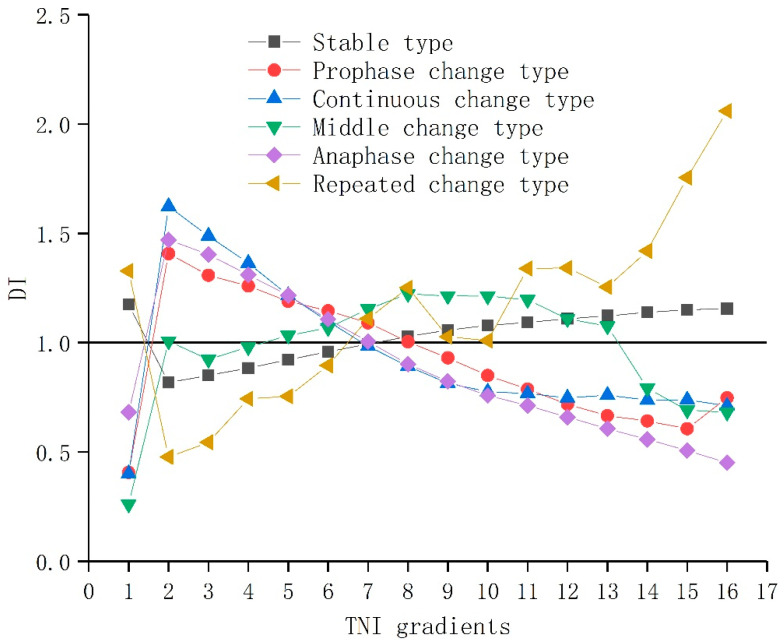
The distribution index of LER change Tupu with different terrain gradients.

**Table 1 ijerph-19-09634-t001:** Transfer matrix for land use in Dianchi Lake basin of China in 1995–2018 (unit: km^2^).

Landscape Type		2018				
		Cultivated Land	Forestland	Grassland	Water Area	Construction Land
1995	Cultivated land	509.614	2.750	0.934	0.149	97.816
	Forestland	0.043	989.286	0.762	0.238	20.148
	Grassland	1.596	4.362	439.344	0.174	38.047
	Water area	--	--	0.007	328.104	1.878
	Construction land	--	0.003	0.698	0.426	468.734

Note: “--” means that no conversion had occurred.

**Table 2 ijerph-19-09634-t002:** Statistics on the area and proportion of LER change Tupu type in the Dianchi Lake basin of China.

LER Change Tupu	Area/km^2^	Ratio/%	Max LER Change Tupu	Area/km^2^	Ratio/%
Stable type	2118.93	73.16	M-M-M-M-M-M	798.91	37.70
Continuous change type	250.62	8.65	ML-ML-ML-L-L-L	120.16	47.95
Anaphase change type	388.85	13.43	MH-MH-MH-MH-M-M	156.25	40.18
Repeated change type	0.98	0.03	M-M-M-MH-MH-M	0.24	24.47
Prophase change type	121.41	4.19	ML-ML-L-L-L-L	48.89	40.27
Middle change type	15.66	0.54	MH-MH-M-MH-MH-MH-MH	4.89	31.24

Note: L (Low), ML (medium low), M (medium), MH (medium high), H (high).

**Table 3 ijerph-19-09634-t003:** The distribution index of LER maximum change Tupu with different terrain gradients.

TNI Gradient	Stable Type		Prophase Change Type		Middle Change Type		Anaphase Change Type		Continuous Change Type		Repeated Change Type	
	MAX Change Tupu	DI	MAX Change Tupu	DI	MAX Change Tupu	DI	MAX Change Tupu	DI	MAX Change Tupu	DI	MAX Change Tupu	DI
1	555555	2.14	433333	0.22	445444	0.65	555544	0.67	222111	0.23	544555	2.36
2	333333	0.42	221111	1.62	443444	0.90	444433	1.00	222111	1.88	455544	0.71
3	333333	0.52	221111	1.47	443444	0.83	444433	1.04	222111	1.70	333443	0.52
4	333333	0.59	221111	1.37	443444	0.91	444433	1.05	222111	1.52	333443	0.64
5	333333	0.69	221111	1.30	443444	0.97	444433	1.07	222111	1.33	333443	0.50
6	333333	0.78	221111	1.22	443444	1.00	444433	1.07	222111	1.14	333443	0.67
7	333333	0.88	221111	1.10	443444	1.14	444433	1.07	222111	0.99	333443	0.93
8	333333	0.99	221111	0.95	443444	1.25	444433	1.06	222111	0.83	333443	1.27
9	333333	1.09	221111	0.83	443444	1.22	444433	1.04	222111	0.70	333443	0.85
10	333333	1.16	221111	0.69	443444	1.08	444433	1.04	222111	0.63	333443	0.78
11	333333	1.22	443333	0.61	443444	1.08	444433	1.02	222111	0.60	333443	1.25
12	333333	1.28	443333	0.63	333433	1.00	444433	0.97	222111	0.57	333443	1.51
13	333333	1.32	443333	0.65	333433	1.05	444433	0.93	222111	0.58	333443	1.09
14	333333	1.38	443333	0.66	333433	0.92	444433	0.82	222111	0.61	333443	0.85
15	333333	1.43	443333	0.57	333433	1.07	444433	0.75	222111	0.66	544555	1.85
16	333333	1.34	443333	0.57	333433	1.52	444433	0.62	222111	0.59	544555	2.64

Note: Low (1), Med-low (2), Medium (3), Med-high (4), High (5).

**Table 4 ijerph-19-09634-t004:** Change in average value of LER of different land use types in the Dianchi Lake basin during 1995–2018.

Year	Cultivated Land	Forestland	Grassland	Water Area	Construction Land
1995	0.2141	0.2080	0.2134	0.2423	0.1859
2000	0.2142	0.2079	0.2133	0.2421	0.1846
2005	0.2137	0.2070	0.2128	0.2427	0.1818
2010	0.2159	0.2060	0.2126	0.2415	0.1715
2015	0.2139	0.2043	0.2106	0.2392	0.1669
2018	0.2134	0.2041	0.2103	0.2391	0.1670
Rate of change (%)	−0.0007	−0.0039	−0.0031	−0.0032	−0.0189

## Data Availability

Not applicable.

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
