# Peer review of "Assessing the Terrain Gradient Effect of Landscape Ecological Risk in the Dianchi Lake Basin of China Using Geo-Information Tupu Method"

_ijerph, 2022, doi:10.3390/ijerph19159634_

Round 1
Reviewer 1 Report
Dear Authors,
Thank you for the very well elaborated work which, compared to other studies, defines in the beginning what is the subject of study properly. The approach is well-devised and structured, which allows you to advance correctly and achieve meaningful results. I have no words to find for any comments and suggestions. Thank you in advance for the kind co-operation!
Kind regards,
Reviewer
Author Response
Thanks for the reviewer's affirmation of the work of this research. I will continue to check and correct necessary mistakes in this study.
Reviewer 2 Report
Dear Authors,
thanks a lot for your manuscript submission. After careful review, my comprehensive evaluation on this paper is a good set of work. The study is fair and convincing, the literal writing is also moderately acceptable. Hence, this research article, can be recommended after minor issues, which I specify mainly in the attached file. Please, for the manuscript preparation follow the MDPI’s rules (https://www.mdpi.com/journal/ijgi/instructions) and accordingly modify references and figures.
I’d like to remark that International Journal of Environmental Research and Public Health (IJERPH) (ISSN 1660-4601) is a peer-reviewed scientific journal which publish papers in the area of environmental health sciences and public health. I think that this part must be stressed in this paper, especially in the discussion and conclusion sections. Finally, if similar studies have been published for the same study area, please mention them in your discussion section and compare you results with their ones.

Author Response
Thanks a lot to reviewers for your comments. I have made modifications according to the reviewer's comments and the parts in the attachment that need to be modified. The following attachment is provide a point-by-point response to the Reviewer's comments.
Kind regards,
author

Reviewer 3 Report
I have checked the paper carefully. In my opinion it can be considered after some major revise n terms of research context and its design. First, the paper is not prepared in journal format and the citations are different from the MDPI format which has to be revised for sure. In addition, the interdiction sections sounds too general to me and there was no a discussion what are he research gap, problem statement, research objective and its contribution as state of art. There are very critical that all are missing. This section also made use of classical citations which has to be updated significantly.
Another major challenge is also with methodology, this section is too short and seems that authors have mixed up methods and results which is quite strange to me.
Author Response
Thanks a lot to reviewers for your comments. I have made modifications according to the reviewer's comments . The following attachment is provide a point-by-point response to the Reviewer's comments.
Kind regards,
author

Reviewer 4 Report
Based on geo- information Tupu method analysis of the terrain gradient effect of landscape ecological risk in the Dianchi Lake basin of China
Using the geoinformation Tupu method, this paper assesses spatio-temporal variations of landscape ecological risk (LER) hoping to improve the environmental conditions of watershed in the Dianchi Lake Basin (DLB) of China. The research findings revealed that since 1995, the landscape of DLB has been changed rapidly from grassland and cultivated land to construction land in the low, med-low, and med-high landscape terrain types. It classifies land by assigning different indices such as landscape fragmentation index (Ci), landscape separation index (Ni) and landscape fractal dimension index (Fi). The paper argues that these indices are influenced by ecosystem services, such as soil conservation, water conservation, carbon storage, water yield, nitrogen and phosphorus output. The authors have observed the changing LER values from 0.213 in 1995 to 0.206 in 2018. This paper recommends optimizing the ecological risk management of medium-risk regions, such as building landscape ecological corridors, lakeside ecological parks, national parks and others.
Overall comments:
The paper has some merits, but in the current form it is not publishable.
First, the writing needs to be totally revised, reorganized, and arguments should be arranged in logical orders.
Second, a series of words like TUPU, prophase, and anaphase are used but without their proper definitions and explanations. It is very difficult to know for what purposes these terms are used and what these words contribute to the scientific community.
Third, the paper is very poorly organized, lengthy, and incoherent.
Though quantitative results and figures try to provide some meaningful information, because of the messy presentation with incoherent arguments throughout the paper, it is not recommended to publish this paper in the current form.
Author Response
Thanks a lot to reviewers for your comments. I have made modifications according to the reviewer's comments. The following attachment is provide a point-by-point response to the Reviewer's comments.
Kind regards,
author

Round 2
Reviewer 3 Report
Thanks for the revise, my comments adressed aready!
Author Response
Dear reviewer,
Thank you for the reviewer's approval of the revision of the comments. I will continue to check and correct necessary mistakes in this study.
Kind regards,
author
Reviewer 4 Report
Page 2: Line 10-12: Assessment of landscape ecological risk (LER) in different terrain gradients is beneficial to ecological environmental protection and risk management.
Page 2: Lines 65-68: Run on sentence.
Page 4, line 134: Awkward writing “Human Computer”??
Page 6: Lines 193-211: Not clear, the paragraph has incomplete information.
Author Response
Dear reviewer 4,
Thanks a lot to reviewer 4 for your comments. I have made modifications according to the reviewer's comments. The following attachment is provide a point-by-point response to the Reviewer's comments.
Kind regards,
author
